# Combined Effect of Biostimulants and Mineral Fertilizers on Crop Performance and Fruit Quality of Watermelon Plants

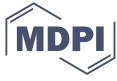

Ângela Fernandes [1,2], Nikolaos Polyzos [3], Filipa Mandim [1,2], Carla Pereira [1,2], Jovana Petrović [4], Marina Soković [4] and Spyridon A. Petropoulos [3,*]

[1] Centro de Investigação de Montanha (CIMO), Instituto Politécnico de Bragança, Campus de Santa Apolónia, 5300-253 Bragança, Portugal; afeitor@ipb.pt (Â.F.); filipamandim@ipb.pt (F.M.); carlap@ipb.pt (C.P.)

[2] Laboratório Associado para a Sustentabilidade e Tecnologia em Regiões de Montanha (SusTEC), Instituto Politécnico de Bragança, Campus de Santa Apolónia, 5300-253 Bragança, Portugal

[3] Department of Agriculture Crop Production and Rural Environment, University of Thessaly, Fytokou Street, 38446 Volos, Greece; npolyzos@uth.gr

[4] Institute for Biological Research "Siniša Stanković"-National Institute of Republic of Serbia, University of Belgrade, Bulevar Despota Stefana 142, 11000 Belgrade, Serbia; jovana0303@ibiss.bg.ac.rs (J.P.); mris@ibiss.bg.ac.rs (M.S.)

[*] Correspondence: spetropoulos@uth.gr; Tel.: +30-2421093196

**Abstract:** The aim of the present study was to evaluate the possible effects of two different biostimulant formulations at different application regimes and combined or not with mineral fertilizers (e.g., W1–W8, including the control treatment (no formulations added)) on the yield parameters and fruit quality of watermelon plants. The highest yield was recorded for the W5 treatment due to the formation of more fruit. The highest content of fat, proteins and ash was recorded for treatment W1, whereas carbohydrates were the most abundant in the control treatment, resulting also in the highest energetic value. The main detected sugars in all the tested samples were sucrose and fructose, which were the highest for the W4 and W5 treatments (sucrose) and W4 treatment (fructose). Malic and citric acid were the most abundant compounds, especially in the W4 treatment. In terms of tocopherols, only α-tocopherol was detected, with the highest amounts being recorded for the W4 treatment. Regarding bioactive properties, the lowest $IC_{50}$ values for OxHLIA were recorded for the W2, W3 and W8 formulations. Moreover, all the extracts exhibited significant anti-inflammatory activity comparable to the positive control, while a variable efficacy of the tested extracts against the studied bacteria and fungi was recorded. In conclusion, our results indicate that simple agronomic practices such as biostimulant application may improve crop performance and improve the proximal composition and the overall quality of watermelon fruit within the context of sustainable crop production.

**Keywords:** *Citrullus lanatus* L.; microminerals; silica; fruit quality; bioactive properties; proximate composition; antioxidant activity; mineral fertilizers





## 1. Introduction

Watermelon (*Citrullus lanatus* L.) belongs to the Cucurbitaceae family, which includes approximately 122 genera and 900 species [1]. The species originates from southern Africa and is an integral part of the Mediterranean diet dating back to 3000 years ago [2]. In Europe, the annual production of watermelon is estimated to be about 5,640,733 tons harvested from approximately 240,994 hectares, whereas in Greece, total production amounts to 430,550 tons in a cultivated area of 8770 hectares [3]. The flesh of watermelon fruit is regarded as the main edible part, but in some areas of the world, people also consume the rinds and the seeds [4]. Nowadays, there is an increasing demand for consumption of healthy foods or products with high added value, which has shifted the market interest

towards the production of functional foods that can complement the human diet [5]. Watermelon has been characterized as a fruit with high antioxidant activity, and it contains a wide range of bioactive compounds [6–8]. According to several studies, watermelon contains a large amount of carotenoids, with lycopene and β-carotene being the main compounds that are responsible for the red color of the flesh, whereas neoxanthin is regarded as the predominant carotenoid for fruit with yellow-colored flesh [9,10]. Lycopene is highly associated with medicinal properties and beneficial health effects against several chronic diseases, namely cancer, gastritis, heart disease, increased blood cholesterol levels and atherosclerosis [11–13]. Similarly, fatty acids (saturated fatty acids—SFA, mounsaturated fatty acids—MUFA and polyunsaturated fatty acids—PUFA) have been associated with the prevention of heart and vascular diseases [14]. In the study carried out by Adetutu et al. [15], the potential health benefits of *C. lanatus* due to its high antioxidant activity were highlighted; authors described its exhibited scavenging efficiency against free radicals. Equally significantly, several studies have pointed out the significant antimicrobial activity of watermelon extracts, which could be further exploited for the control of various bacteria, namely *Escherichia coli*, *Klebsiella pneumonia*, *Bacillus subtilis*, *B. pumilus*, *Salmonella typhi*, *Enterococus faecalis*, *Vibrio cholerae*, *Shigella dysenteriae*, *Proteus mirabilis* and *P. vulgaris*; and fungi such as *Candida albicans*, *Aspergilus niger*, *A. flavus*, *Penicillium chrysogenum*, *P. notatum*, *Trichosporon bepelli* and *Trichophyton mentagraphytes* [16–20].

The increasing demands of producing the food needed to ensure food security for the ever-growing world population, combined with the ongoing climate crisis and land degradation, make it necessary to redesign modern horticulture and adopt innovative agronomic practices that reduce environmental impacts and preserve agrobiodiversity and natural resources without compromising the yield of crops [21–24]. The use of plant biostimulants is a novel and promising agronomic tool that can help farmers to face the modern challenges of crop production by increasing crop yield, nutrients use efficiency and the quality of vegetables under biotic and abiotic stressors [25–27]. Of particular interest is the use of biostimulants in horticultural crops due to the sensitivity of such crops to stressors, as well as to the intensification of production systems [28–31]. Due to the growing interest in integrating plant biostimulants in modern agriculture systems, several studies have been carried out in order to investigate their possible effects on Cucurbitaceae plants in terms of growth and development, crop yield, tolerance to biotic and abiotic stressors, nutrient use efficiency, and mineral and chemical composition [32–36]. For example, Bijalwan et al. [37], who studied the effect of mycorrhiza (*Glomus mosseae* and *Gigaspora gigantean*) and silicon application on watermelon plants grown under saline conditions reported that silicon treatments increased plant growth and fruit yield and decreased salt-related oxidative stress. On the other hand, Toresano-Sánchez et al. [38] suggested that silicon application did not increase total soluble solids and ascorbic acid content, while no differences regarding the peel and pulp color were reported compared to untreated plants. In the same line, de Paula et al. [27] did not detect a significant effect on watermelon yield parameters (total yield, number of fruit, fruit mean weight) after the pre-harvest application of a biostimulant formulation consisting of agave extracts and microminerals with hormonal activities, whereas a negative effect on soluble solids content was reported. The use of microbial biostimulants may have beneficial effects on both soil functions and watermelon plant growth, as reported by Peng et al. [39], who applied two microbial agents (*Paecilomyces lilacinus* and *Bacillus subtilis*) in a watermelon monocropping system; or Ning et al. [40], who tested the application of a biofertilizer based on *Paenibacillus polymyxa*. Similarly, the use of seaweed extracts (*Ascophyllum nodosum*) also improved the crop yield of watermelon plants, whereas a variable effect on quality parameters was suggested depending on the biostimulant application regime and dose. The same type of biostimulants also had a positive effect on alleviating the negative impact of drought stress on watermelon transplants [41]. On the other hand, Soteriou et al. [42] suggested that the application of protein hydrolysates did not affect yield parameters, and

most of the fruit quality features assessed in grafted watermelon plants were due to the high vigor of the rootstock, which concealed any impacts of the biostimulant formulation.

Despite the high number of reports regarding the use of biostimulants on various vegetable species, there is scarce literature regarding the effects of biostimulants on watermelon crop performance and the quality of fruit. Therefore, the current study aimed to evaluate the effects of different biostimulant formulations and mineral fertilizers on yield parameters and fruit quality related to the proximal composition, chemical composition, and bioactive properties of the watermelon grown and determine whether its application is beneficial and can improve the overall quality and yield of fruit.

## 2. Materials and Methods

### 2.1. Plant Material and Growing Conditions

The trial was carried out at the experimental field of the University of Thessaly in the Velestino region during the growing period of May–July 2019. Seedlings of watermelon scions (*Citrullus lanatus* L.; Arena $F_1$) grafted onto Vitalley $F_1$ rootstocks were bought by Elanco Hellas (Greece) and transplanted at the field on 15 May, when they reached the stage of 3–5 true leaves. The total experimental area was approximately 2900 m$^2$, while the plants were planted in rows with distances of 2.5 m between the rows and 1.6 m within the rows (plant density of 2500 plants per acre). The plants were planted in a total of 24 rows using 3 rows per treatment randomly distributed in the experimental area, while 30 plants were planted in each row, for a total of 720 plants. Each row was covered with biodegradable plastic film (linear low-density polyethylene (LLDP) with thickness of 50 μm) to inhibit the growth of weeds. The soil was sandy clay loam (38% sand, 36% silt, and 26% clay), with pH = 7.4 (1:1 soil/H$_2$O), organic matter content = 1.3%, CaCO$_3$ = 10.4%, and organic carbon content 1.7%. Soil properties were determined in three samples collected from different sites of the experimental field. Prior to transplantation, a base dressing with 250 kg/ha of mineral fertilizer (15-15-15) was applied on 8 May and was evenly distributed with a rototiller. After transplantation, fertilization was applied via side dressings or the drip irrigation system (fertigation). In particular, on 21 May (6 days after transplantation; DAT) plants were supplied with phosphoric humic acids at 10 L/ha via fertigation; on 10 June (26 DAT), a side dressing was applied with 50 kg/ha of 30-10-10 (N-P-K); on the 12 (28 DAT) and 14 (30 DAT) of June, ammonium nitrate fertilizer (34.5-0-0; N-P-K) at 70 kg/ha was implemented. Moreover, on 19 June (35 DAT), plants were supplied via drip irrigation system (2 kg/ha) with a fruiting–setting biofertilizer (Disper Bloom; Agrofarm S.A., Greece) that contained 0-4.3-4.8 (N-P-K), 26.3% seaweed extracts, 8.0% free amino acids, 1.56% Zn-EDTA, 0.75% B, 0.26% Mo, 5.50% polysaccharides, 0.30% vitamins of B complex; on 24 June (40 DAT) one more fertilization was applied with 50 kg/ha (20-20-20; N-P-K) at fruit formation stage. The total amounts of nutrients for all the treatments were 110.8-52.6-52.6 kg/ha of N-P-K. Irrigation was applied via a drip irrigation system (one emitter per plant with water supply of 4 L/h) at regular intervals based on the environmental conditions during the experimental period. Weeds were controlled chemically with pre-emergence herbicides, as well as manually with a hand hoe during the growing period and until crop establishment. Pests and pathogens were chemically controlled based on the recommended practices of the crop. Harvest took place when fruit reached a marketable size, namely on the 22 and 24 of July (66 DAT and 68 DAT, respectively). Figure 1 presents different developmental stages of plants and fruit before and after harvest.

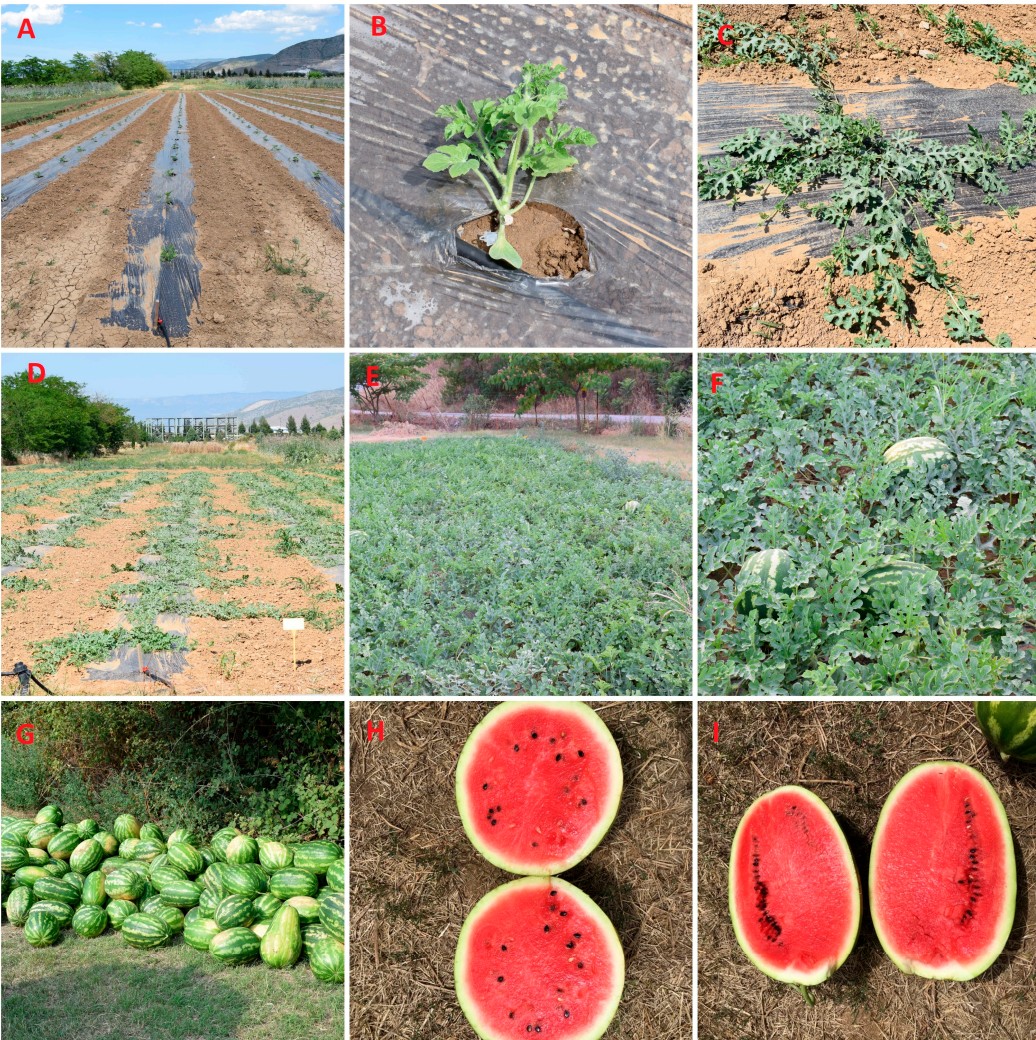

**Figure 1.** Pictures of the plants throughout the experiment. (**A**) Field after plant transplantation. Rows of plants were covered with plastic film. (**B**) Plant directly after transplantation. (**C–F**) Plants at different growth stages until fruit maturation. (**G**) Collected fruit. (**H**) Equatorial section of fruit. (**I**) Longitudinal section of fruit. Photos are from the personal record of Spyridon A. Petropoulos.

## 2.2. Experimental Treatments

The tested formulations are presented in Table 1, including five different products that contained $SiO_2$; $SiO_2$ and CaO; a mixture of natural metabolic catalysts, amino acids and trace elements; mineral fertilizer of 15-65-15 (N-P-K) and trace elements; and mineral fertilizer of 15-10-50 (N-P-K) and trace elements. The experiment consisted of eight treatments including the control treatment, and the application dose and schedule are presented in detail in Table 2. All the tested formulations were applied via the irrigation system (direct soil application).

**Table 1.** The detailed composition of the tested biostimulant formulations.

| Formulation | Composition |
|---|---|
| A | 92% $SiO_2$ *w/w* |
| B | 35% *w/v* $SiO_2$ + 35% *w/v* CaO |
| C | Mixture of natural metabolic catalysts, amino acids, and trace elements |
| D | 15-65-15 (N-P-K) + trace elements |
| E | 15-10-50 (N-P-K) + 3MgO + trace elements |

**Table 2.** The experimental treatments of the study.

| Treatment | Application | Dose | Time Schedule |
|-----------|-------------|------|---------------|
| W1 | One application of formulation A | 20 kg/ha/application | 15 DAT * |
| W2 | One application of formulation B | 20 L/ha/application | 15 DAT |
| W3 | Three applications of formulation A | 20 kg/application | 15 DAT<br>35 DAT<br>55 DAT |
| W4 | Three applications of formulation B | 20 L/ha/application | 15 DAT<br>35 DAT<br>55 DAT |
| W5 | Three applications of formulation B | 20 L/ha/application (B) | 15 DAT<br>35 DAT<br>55 DAT |
| | One application of formulation C | 10 L/ha (C) | 15 DAT<br>At flowering initiation |
| | Two applications of formulation D | 30 L/ha/application | At fruit setting |
| | Three applications of formulation E | 30 L/ha/application | At fruit enlargement and two more applications at 10-day intervals after the first application |
| W6 | Six applications of formulation A | 20 kg/application | 15 DAT<br>25 DAT<br>35 DAT<br>45 DAT<br>55 DAT<br>65 DAT |
| W7 | Six applications of formulation B | 20 L/application | 15 DAT<br>25 DAT<br>35 DAT<br>45 DAT<br>55 DAT<br>65 DAT |
| W8 | Control | - | - |

* DAT: days after transplantation.

### 2.3. Fruit Quality Parameters

After harvest, the total number of fruits per treatment was recorded, while fruit dimensions (perimeter 1 (P1): longitudinal circumference) and perimeter 2 (P2): equatorial circumference) were determined with a tape measure and expressed in cm. Total yield was calculated by adding the weight of individual fruit and expressed in kg/ha. The thickness of fruit rind (cm) was determined at the equatorial axis with a caliper after cutting fruit with a knife and removing the edible part (flesh). For fruit quality assessment, flesh color and firmness, pH, electrical conductivity (EC), titratable acidity (TA) and total soluble solids (TSS) of fruit juice were recorded in ten randomly selected fruits from each treatment, as described by Petropoulos et al. [43]. In brief, flesh color was determined at three distinct points in the middle of fruit flesh with a chromameter (CR 400; Konica Minolta, Tokyo, Japan). Color was assessed according to the International Commission on Illumination (CIE) standard, using the measurement space of the CIE, $L^*$ $a^*$ $b^*$, where $L^*$ represents the luminosity ($L = 0$ black, $L = 100$ white), $a^*$ represents the redness ($-a = 0$ greenness, $+a$ = redness) and $b^*$ represents yellowing ($-b$ = bluish; $+b$ = yellowish). For chroma ($C^*$) and hue ($h^*$) values, the following formulas were used [44]:

$$C^* = \sqrt{a^{*2} + b^{*2}}$$

$$h^* = 180 + \left( \frac{\left( arctan \frac{b^*}{a^*} \right)}{6.2832} \right) * 360, \; when \; a^* < 0 \; and$$

$$h^* = \left( \frac{\left( arctan \frac{b^*}{a^*} \right)}{6.2832} \right) * 360, \; when \; a^* > 0$$

Flesh firmness was assessed manually with a penetrometer after removing a cubic section (5 cm × 5 cm) from the middle of fruit flesh using a penetrometer (TR53205; T.R. Turoni, SRL, Italy) with a pointed probe forced to a depth of 8 mm and was expressed in newtons (N) [43]. The same cubic portion of flesh was ground with a domestic electric blender and squeezed with a tool to collect the juice for TSS, TA, EC and pH determination. Total soluble solids were measured with a hand-held refractometer (TR53000C; T.R. Turoni SRL, Forli, Italy) and expressed in ° Brix [43]. Electrical conductivity and pH of juice were determined with a portable EC (MW 302 PRO EC Meter; Milwaukee Instruments Inc., Rocky Mount, NC, USA) and pH meter (MW101 PRO pH Meter; Milwaukee Instruments Inc., Rocky Mount, NC, USA), respectively [43]. Titratable acidity was determined after titration with 0.02 M NaOH to pH 8.1 and was expressed as g of citric acid per 100 mL of juice [43].

### 2.4. Chemical Analysis

#### 2.4.1. Proximate Composition and Energetic Value

Lyophilized fruit samples were analyzed for proximal composition (carbohydrates, fat, proteins and ashes) according to the Official Methods of Analysis of the AOAC [45]. Crude protein was analyzed using the macro-Kjeldahl method (N × 6.25) with an automatic distillation and titration unit (model Pro-Nitro-A, JP Selecta, Abrera, Barcelona). Incineration at 550 ± 15 °C was used to assess the ash content. Crude fat was determined using a Soxhlet apparatus by extracting a known weight of the sample with petroleum ether and the results were expressed in g/100 g of dry weight (dw). Total carbohydrates were calculated by difference: total carbohydrates (g/100 g dw) = 100 − (g protein + g fat + g ash). The energetic value was calculated according to the Atwater system using this formula: energy (kcal/100 g dw) = 4 × (g protein + g carbohydrates) + 9 × (g fat).

#### 2.4.2. Hydrophilic Compounds
Sugars

After extraction, free sugars were determined by high-performance liquid chromatography coupled to a refraction index detector (HPLC-RI) (HPLC-RI; Knauer, Smartline 1000 and RI-Knauer, Berlin, Germany, respectively), operating as previously described by Spréa et al. [46]. Identification was performed by comparison of their relative retention times (Rt) with authentic standards (Sigma-Aldrich, St. Louis, MO, USA), and quantification using melezitose as internal standard (IS, Sigma-Aldrich, St. Louis, MO, USA). The results were recorded and processed using Clarity 2.4 software (DataApex, Podohradska, Czech Republic) and expressed in g/100 g of dw.

Organic Acids

Organic acids were determined following the procedure previously described by the authors [47]. The evaluation was performed by ultra-fast liquid chromatography coupled with the photodiode array detector (UFLC-PDA; Shimadzu Corporation, Kyoto, Japan), and the chromatographic separation occurred in a C18 SphereClone (Phenomenex, Torrance, CA, USA) reverse phase column (5 μm, 250 × 4.6 mm i.d.) thermostated at 35 °C, using 3.6 mM sulfuric acid solution as an eluent at a flow rate of 0.8 mL/min. The identification was carried out by comparing the chromatograms obtained for the analyzed samples with commercial standards. The quantification of organic acids was done by relating the peak

areas, recorded at 215 nm, with the calibration curves obtained with authentic commercial standards (Sigma-Aldrich, St. Louis, MO, USA) for each compound. The results are expressed in g/100 g dw.

### 2.4.3. Lipophilic Compounds
### Fatty Acids

Fatty acid methyl esters (FAME) were investigated after trans-esterification of the lipid fraction obtained through Soxhlet extraction as previously described by Spréa et al. [47], and determined by gas–liquid chromatography with flame ionization detection, using a YOUNG IN Chromass 6500 GC System instrument equipped with a *split/splitless* injector, a flame ionization detector (FID) and a Zebron-Fame column. Fatty acids identification and quantification were performed by comparing the relative retention times of FAME peaks from samples with standards (standard mixture 47885-U, Sigma, St. Louis, MO, USA) and results were recorded and processed using the Software Clarity DataApex 4.0 Software (Prague, Czech Republic) and expressed in relative percentages of each fatty acid.

### Tocopherols

Tocopherols were determined following a procedure previously described by Sprea et al. [47]. The formerly described HPLC system, coupled to a fluorescence detector (FP-2020; Jasco, Tokyo, Japan) programmed for excitation at 290 nm and emission at 330 nm was used. The compounds were identified by chromatographic comparisons with authentic commercial standards (Matreya, Pleasant Gap, PA, USA), and quantification was based on the response of the fluorescence signal, using the internal standard method and by chromatographic comparison with standards. Tocol (Matreya, Pleasant Gap, PA, USA) was used as an internal standard, and the results were expressed in µg/100 g dw.

### 2.4.4. Carotenoid and Chlorophylls Content

The content of carotenoids and chlorophylls was evaluated using a method described by the authors [48]. Briefly, the samples (500 mg) were vigorously shaken with 10 mL of acetone/hexane mixture (4:6, *v/v*) for 1 min and filtered through Whatman No. 4 filter paper. The absorbance was measured at 453, 505, 645 and 663 nm, and the contents of carotenoids (β-carotene and lycopene) and chlorophyll a and b were obtained with the following equations, and expressed in mg per 100 g of dw: β-carotene (mg/100 mL) = 0.216 × A663 − 1.220 × A645 − 0.304 × A505 + 0.452 × A453; lycopene (mg/100 mL) = −0.0458 × A663 + 0.204 × A645 − 0.304 × A505 + 0.452 × A453; chlorophyll a (mg/100 mL) = 0.999 × A663 − 0.0989 × A645; chlorophyll b (mg/100 mL) = −0.328 × A663 + 1.77 × A645.

### *2.5. Bioactive Properties*
### 2.5.1. Antioxidant Activity

The antioxidant activity of the hydroethanolic extracts was determined by one ex vivo assay, namely, the inhibition of oxidative hemolysis (OxHLIA) in sheep's blood erythrocytes, as described by Lockowandt et al. [49]. The $EC_{50}$ values (sample concentration providing 50% of antioxidant activity or 50% hemolysis inhibition for $\Delta t$ of 60 min, $IC_{50}$; µg/mL) were used to express the results. Trolox (Sigma-Aldrich, St. Louis, MO, USA) was used as a positive control.

### 2.5.2. Cytotoxicity Assays

The hydroethanolic extracts' cytotoxicity was assessed by the sulforhodamine B assay against four human tumor cell lines: AGS (gastric adenocarcinoma), colon adenocarcinoma (Caco-2), MCF-7 (breast adenocarcinoma) and NCI-H460 (non-small cell lung cancer). Cell lines were seeded in 96-well plates, with a final density of $1.0 \times 10^4$ cells/well, and fixation was allowed for 24 h. Subsequently, various concentrations of the extract were added to the cells and incubated for 48 h, as described by Sprea et al. [47]. Ellipticine (Sigma-Aldrich, St. Louis, MO, USA) was used as a positive control. The same assay was

also used to determine the hepatotoxicity of the extracts against a non-tumor African green monkey kidney cell line (Vero). Results were expressed as $GI_{50}$ values, relative to the extract concentration ($\mu g/mL$) that caused 50% of cell growth inhibition.

### 2.5.3. Anti-Inflammatory Activity

The anti-inflammatory activity of the hydroethanolic extracts (400 to 6.25 $\mu g/mL$) was evaluated based on nitric oxide (NO) production in a RAW 264.7 murine macrophage cell line (ECACC 91062702) due to lipopolysaccharide (LPS, 1 mg/mL in DMEM; Sigma-Aldrich, Saint Louis, MO, USA) stimulation. For that purpose, the Griess Reagent System kit (Promega, Madison, WI, USA) was used as described in previously published protocols [50]. Dexamethasone (50 mM) (Sigma-Aldrich, Saint Louis, MO, USA) and samples without the addition of LPS were used as positive and negative controls, respectively. The NO generated was monitored at 540 nm (ELX800 Biotek microplate reader; Bio-Tek Instruments Inc., Winooski, VT, USA). The extract concentrations that triggered 50% of NO production inhibition were expressed as $EC_{50}$ values ($\mu g/mL$).

### 2.5.4. Antimicrobial Activity

Three Gram-positive bacteria (*Staphylococcus aureus* (ATCC 11632), *Bacillus cereus* (clinical isolate) and *Listeria monocytogenes* (NCTC 7973)), and three Gram-negative bacteria (*Salmonella enterica* subsp. *enterica serovar typhimurium* (ATCC 13311), *Escherichia coli* (ATCC 25922), and *Enterobacter cloacae* (ATCC 35030)) were selected to test the antibacterial activity of the hydroethanolic extracts.

For antifungal activity, six micromycetes were used, namely, *Aspergillus fumigatus* (ATCC 9197), *Aspergillus ochraceus* (ATCC 12066), *Aspergillus niger* (ATCC 6275), *Penicillium funiculosum* (ATCC 36839), *Penicillium verrucosum* var. *cyclopium* (food isolate) and *Trichoderma viride* (IAM 5061).

The microdilution method to determine the minimum inhibitory, bactericidal and fungicidal concentrations (MICs, MBCs and MFCs, expressed in mg per mL) was performed as previously described by Soković et al. [51]. The food preservatives sodium sulfite (E221) and potassium metabisulphite (E224) (Sigma-Aldrich, St. Louis, MO, USA) were used as positive controls.

### 2.6. Statistical Analysis

The field experiment was laid out according to randomized complete block (RCB) design with three replications (three rows of 30 plants each, for each treatment as described in detail in Section 2.2 and Tables 1 and 2). Chemical composition and bioactivity assays were performed in three different samples for each treatment, whereas all analyses were performed in triplicate. The results are presented as mean values and standard deviations (SD). All data were checked for normal distribution according to the Shapiro–Wilk method and then analyzed with one-way ANOVA. The mean values were compared by Tukey's honest significant difference (HSD) test at $p < 0.05$. All statistical analyses were processed with Statgraphics 5.1. plus (Statpoint Technologies, Inc., Warrenton, VA, USA).

In addition, a principal component analysis (PCA) was performed to evaluate the contribution of each variable to the total diversity and classify the tested biostimulant formulations according to crop performance (yield), proximal and chemical composition, and bioactive properties. This analysis was carried out with the statistical software Statgraphics 5.1. plus (Statpoint Technologies, Inc., Warrenton, VA, USA).

### 3. Results and Discussion

#### 3.1. Yield Parameters

Table 3 presents the results regarding the crop performance of watermelon plants. The combined application of biostimulants (e.g., $SiO_2$, CaO, natural metabolic catalysts, amino acids, and trace elements) and mineral fertilizers (W5) resulted in higher fruit yield per hectare (84,042 kg/ha), which was mostly associated with the highest fruit setting

(9930 no of fruit/ha), since the highest fruit weight was recorded for the W1 and W2 treatments (one application of formulations that contained $SiO_2$ (92%, *w/w*), $SiO_2$ (35% *w/v*) and CaO (35% *w/v*), respectively. Moreover, the W5 and W1 treatments significantly increased the longitudinal perimeter (e.g., fruit length; 96.5 and 96.48 cm, respectively), while the equatorial perimeter was the highest for the W5 treatment (79.12 cm). Similarly, the highest value of the P1/P2 ratio was recorded for the W1 treatment (1.24), indicating that fruit dimensions can be slightly regulated by simple agronomic practices such as biostimulant application.

**Table 3.** Yield parameters of watermelon plants in relation to biostimulant application.

| Treatment | Yield (kg/ha) | No of Fruit/ha | Mean Fruit Weight (kg) | Fruit Perimeter 1 (P1) (cm) | Fruit Perimeter 2 (P2) (cm) | P1/P2 | Rind Thicness (cm) |
|---|---|---|---|---|---|---|---|
| W1 | 78,226 ± 1523 c | 8299 ± 235 d | 9.43 ± 0.42 a | 96.5 ± 4.3 a | 77.3 ± 3.5 b | 1.24 ± 0.03 a | 1.23 ± 0.13 d |
| W2 | 76,555 ± 1274 d | 8299 ± 367 d | 9.20 ± 0.76 a | 91.48 ± 5.27 c | 75.38 ± 4.61 d | 1.21 ± 0.05 b | 1.33 ± 0.11 c |
| W3 | 81,521 ± 897 b | 9479 ± 668 b | 8.55 ± 1.08 c | 89.7 ± 4.11 d | 75.6 ± 3.1 d | 1.19 ± 0.02 c | 1.27 ± 0.22 cd |
| W4 | 69,455 ± 1158 f | 7812 ± 777 e | 8.85 ± 0.54 b | 95 ± 4 b | 77.7 ± 3.2 b | 1.21 ± 0.05 b | 1.52 ± 0.28 a |
| W5 | 84,042 ± 1897 a | 9930 ± 395 a | 8.47 ± 0.85 c | 96.48 ± 3.97 a | 79.12 ± 2.78 a | 1.22 ± 0.03 b | 1.42 ± 0.21 b |
| W6 | 75,226 ± 956 e | 8785 ± 459 c | 8.55 ± 1.09 c | 87.84 ± 7.05 e | 75.22 ± 3.43 d | 0.86 ± 0.10 d | 1.17 ± 0.12 e |
| W7 | 74,267 ± 1101 e | 8368 ± 332 d | 8.67 ± 1.55 c | 89.04 ± 5.61 d | 76.05 ± 8.69 c | 1.19 ± 0.09 c | 1.42 ± 0.18 b |
| W8 | 76,090 ± 1024 d | 8333 ±551 d | 8.64 ± 2.02 c | 94.49 ± 6.41 b | 77.82 ± 3.71 b | 1.21 ± 0.06 b | 1.33 ± 0.12 c |

Mean values of the same column followed by different letters are significantly different according to Tukey's honest significant difference (HSD) test at $p < 0.05$. The description of treatments is presented in detail in Table 2.

Regarding rind thickness, the W4 treatment (three applications of 35% *w/v* $SiO_2$ and 35% *w/v* CaO) showed the highest value (1.52 cm), being significantly different from the rest of the treatments. This finding is of particular importance for the post-harvest behavior of watermelon fruit, since rind thickness is associated with a better shelf life of fruit and facilitates better transportation of fruit throughout the marketing chain [52]. Similarly to our study, da Luz Neto et al. [53] reported that foliar application of Si on mini watermelons resulted in an increase in peel thickness. Based on literature reports, a varied effect of biostimulant application on the crop performance of watermelon was expected, depending on the biostimulant formulation. For example, the use of biofertilizers that contained growth-promoting rhizobacteria increased the yield of cucumber [54], while similar results were suggested for two microbial agents (*Paecilomyces lilacinus* and *Bacillus subtilis*) for watermelon. Regarding the silica-based biostimulants, Cristofano et al. [23] classified plant species as high and intermediate accumulators of silica or as silica excluders and further suggested that plant responses to silica application and the mitigation of stressor effects depends on this classification. The positive effects of silica observed in our study could be associated with the improvement of water balance through the upregulation of silicon-transporting aquaporins and the increased accumulation of various compounds with osmoprotective properties [55,56]. Moreover, plants treated with silicon may develop a higher root biomass and root surface that facilitates their improved uptake of the available water. In contrast, according to Toresano et al. [57], no significant effects of silicon were observed on the yield parameters of watermelon fruit. Apart from silicon, the additional amounts of fertilizers provided through the applications of formulations D and E implemented in our study, as well as the natural metabolic catalysts, amino acids and trace elements that the W5 treatment included, could be responsible for the beneficial effects of this treatment on fruit yield. Regarding fruit dimensions, the literature reports also indicate the impact that silicon application may have, as, for example, in the case of strawberry (*Fragaria × ananasa* Duch. var. Fortuna) [58] or squash fruit (*Cucurbita pepo* L.) [59]. However, considering that in our study a varied effect was recorded, the application dose and the inclusion of other compounds in biostimulant formulations (e.g., mineral fertilizers, trace elements, amino acids and natural metabolic catalysts) could be also involved in the regulation of fruit dimensions and rind thickness.

### 3.2. Fruit Quality Parameters

The results of color parameter analysis are presented in Table 4. The application of the tested formulation resulted in significant differences in specific color parameters of fruit flesh, whereas $L*$ and $b*$ mean values of the treatments did not differ from each other. In particular, the W4 treatment (three applications of 35% ($w/v$) $SiO_2$ and 35% ($w/v$) CaO) resulted in the highest $a*$ (more red flesh; 23.65) and $C*$ (more intense color; 25.71) values, being significantly different only from the W1 treatment (one application of 92% ($w/w$) $SiO_2$). On the other hand, the W6 treatment (six applications of 92% ($w/w$) $SiO_2$) recorded the highest $h*$ values (24.17), being significantly different from the W4 treatment. In contrast to our study, Peris-Felipo et al. [58] did not observe any significant effects of silicon application on the color of strawberry fruit, whereas Karagiannis et al. [60] reported a varied effect on the color of apple (*Malus domestica* L. Borkh) peels. On the other hand, according to González-Terán et al. [61], silicon application may significantly increase chroma values of cucumber fruit (*Cucumis sativus* L. var. Modan). Similar results to our study were also recorded for the impact of potassium silicate application on the peel and flesh color of peach and nectarine fruit (*Prunus persica* (L.) Batsch), with only specific parameters being affected by the treatments, although these effects were attributed to potassium properties rather than silicon itself [62].

**Table 4.** Flesh color parameters in relation to biostimulant formulations.

| Treatment | Flesh Color | | | | |
|---|---|---|---|---|---|
| | $L*$ | $a*$ | $b*$ | $C*$ | $h*$ |
| W1 | 30.2 ± 1.7 a | 21.55 ± 1.65 b | 9.59 ± 0.72 a | 23.59 ± 1.79 b | 23.99 ± 0.43 a |
| W2 | 31.88 ± 4.16 a | 22.48 ± 2.59 ab | 9.97 ± 1.01 a | 24.59 ± 2.75 ab | 23.96 ± 0.92 a |
| W3 | 31.81 ± 2.53 a | 22.55 ± 1.53 ab | 9.93 ± 0.55 a | 24.64 ± 1.61 ab | 23.79 ± 0.59 a |
| W4 | 30.49 ± 3.44 a | 23.65 ± 1.74 a | 10.10 ± 0.9 a | 25.71 ± 1.95 a | 23.10 ± 0.65 b |
| W5 | 31.66 ± 2.35 a | 21.95 ± 1.58 ab | 9.74 ± 0.77 a | 24.02 ± 1.73 ab | 23.92 ± 0.68 a |
| W6 | 31.3 ± 2.4 | 22.5 ± 2.9 ab | 10.10 ± 1.17 a | 24.69 ± 3.11 ab | 24.17 ± 0.57 a |
| W7 | 31.56 ± 4.02 a | 22.21 ± 2.82 ab | 9.94 ± 1.38 a | 24.33 ± 3.12 ab | 24.08 ± 0.82 a |
| W8 | 30.43 ± 3.01 a | 22.75 ± 2.59 ab | 10 ± 1 a | 24.85 ± 2.78 ab | 23.8 ± 0.7 a |

Mean values of the same column followed by different letters are significantly different according to Tukey's honest significant difference (HSD) test at $p < 0.05$. The description of treatments is presented in detail in Table 2. $L*$: represents the luminosity ($L = 0$ black, $L = 100$ white); $a*$: represents the redness ($-a = 0$ greenness, $+a$ = redness); $b*$: represents yellowing ($-b$ = bluish; $+b$ = yellowish).

Regarding the determined fruit quality parameters, the applied biostimulant formulations did not affect the firmness of flesh (Table 5). In contrast, the W2 and W8 treatments resulted in the highest total soluble solids content and pH values (12.56° Brix and 5.51, respectively); although they differed significantly only from the W6 and W4 treatments, respectively. Moreover, the highest EC values were recorded for the W7 treatment (2.59 mS/cm), being significantly different from treatments W1 and W2, while the control treatment had the highest citric acid content (e.g., titratable acidity; 0.11 g citric acid/100 mL of flesh juice), without observation of significant differences from the W4, W5 and W7 treatments. Considering the importance of the determined parameters on taste and flavor perception [63], it could be suggested that the tested formulation may affect fruit quality; however, no significant improvement compared to the control treatment was recorded for any of the biostimulant treatments, except for the case of EC values and titratable acidity, where specific biostimulant formulations improved these parameters compared to the control. In other studies where the effect of biostimulants on fruit quality was evaluated, a varied impact was recorded. For example, Soteriou et al. [42] did not record a significant effect of protein hydrolysates on either soluble solids content or titratable acidity of watermelon fruit, while similar results have been suggested for the titratable acidity of other fruit, such as tomato (*Solanum lycopersicum* L.) and melon (*Cucumis melo* L.) [64,65]. On the other hand, Salim et al. [59] reported that Si application may improve total soluble sugar content in squash leaves under either normal or deficit irrigation, since Si is associated with tolerance to abiotic stressors and its application induces osmoprotective mechanisms.

Moreover, Nasrallah et al. [66] suggested that Si application may induce the accumulation of metabolites and the direct modification of starch to soluble sugars. In the same line, Emad et al. [67] suggested a varied effect of silicon on the total soluble solids of cucumber fruit depending on the dose and the application method (foliar or soil application) of the biostimulant and the harvesting date, whereas Costan et al. [68] did not record any effects of Si application on tomato fruit acidity. Interestingly, no significant effect on fruit firmness was recorded for the tested biostimulants in our study, since, according to the literature, Si application is expected to increase fruit firmness due to its involvement in cell wall structure and the induced biosynthesis of lignin and hemicellulose [69,70]. However, our results are in agreement with the study of Soteriou et al. [42], who also did not record any significant effect of biostimulant application on the firmness of watermelon fruit flesh, which highlights the impacts the application dose and biostimulant composition may have on fruit quality parameters.

**Table 5.** Fruit quality parameters in relation to biostimulant formulations.

| Treatment | Flesh Firmness (N) | TSS (° Brix) | pH | EC (mS/cm) | Titratable Acidity (g of Citric Acid/100 mL of Juice) |
|---|---|---|---|---|---|
| W1 | $-0.49 \pm 0.01$ a | $12.49 \pm 0.67$ a | $5.49 \pm 0.09$ a | $2.34 \pm 0.14$ b | $0.09 \pm 0.01$ bc |
| W2 | $-0.48 \pm 0.08$ a | $12.56 \pm 0.72$ a | $5.41 \pm 0.09$ ab | $2.34 \pm 0.07$ b | $0.09 \pm 0.01$ bc |
| W3 | $-0.55 \pm 0.12$ a | $12.22 \pm 0.41$ a | $5.46 \pm 0.13$ ab | $2.43 \pm 0.14$ ab | $0.09 \pm 0.01$ bc |
| W4 | $-0.48 \pm 0.15$ a | $12.09 \pm 0.78$ ab | $5.38 \pm 0.08$ b | $2.40 \pm 0.18$ ab | $0.10 \pm 0.01$ ab |
| W5 | $-0.46 \pm 0.09$ a | $12.31 \pm 0.83$ a | $5.50 \pm 0.14$ a | $2.53 \pm 0.06$ a | $0.10 \pm 0.01$ ab |
| W6 | $-0.52 \pm 0.13$ a | $11.60 \pm 0.82$ b | $5.47 \pm 0.06$ ab | $2.53 \pm 0.09$ ab | $0.08 \pm 0.02$ c |
| W7 | $-0.55 \pm 0.15$ a | $12.15 \pm 0.42$ a | $5.48 \pm 0.09$ ab | $2.59 \pm 0.17$ a | $0.10 \pm 0.01$ ab |
| W8 | $-0.46 \pm 0.06$ a | $12.08 \pm 0.43$ ab | $5.51 \pm 0.07$ a | $2.55 \pm 0.33$ a | $0.11 \pm 0.02$ a |

Mean values of the same column followed by different letters are significantly different according to Tukey's honest significant difference (HSD) test at $p < 0.05$. The description of treatments is presented in detail in Table 2.

### 3.3. Chemical Composition

#### 3.3.1. Proximate Composition and Energetic Value

The effect of biostimulant formulation application on the proximal composition of watermelon fruit is presented in Table 6. The highest content of fat, proteins and ash was recorded for treatment W1 (0.46 g/100 g dw, 7.36 g/100 g dw and 3.42 g/100 g dw, respectively), where 92% (*w/w*) of $SiO_2$ was applied only once, whereas carbohydrates were the most abundant in the control treatment (W8; 92.7 g/100 g dw), resulting also in the highest energetic value (382.2 Kcal/100 g dw), although no significant differences were recorded between the W8 and W4 treatments in terms of the energy content of the fruit. To the best of our knowledge, this is the first report regarding the effects of Si-based biostimulants on the proximal composition of watermelon fruit. Based on other literature reports, biostimulant application may regulate the proximal composition of horticultural products such as garlic (*Allium sativum* L.) [71], common bean (*Phaseolus vulgaris* L.) [30], and tomato [27,72], among others, whereas it is suggested there is a high dependency on biostimulant composition. These findings indicate that further research is needed to reveal the impact of biostimulant formulations that will improve the values of the quality parameters of watermelon fruit.

**Table 6.** Proximal, energetic value and hydrophilic compounds of the studied watermelon samples (mean $\pm$ SD, $n$ = 3) in relation to biostimulant formulation.

| | W1 | W2 | W3 | W4 | W5 | W6 | W7 | W8 |
|---|---|---|---|---|---|---|---|---|
| **Proximal Composition** | | | | (g/100 g dw) | | | | |
| Fat | 0.46 ± 0.01 a | 0.36 ± 0.01 c | 0.31 ± 0.02 d | 0.25 ± 0.01 f | 0.42 ± 0.01 b | 0.34 ± 0.01 c | 0.28 ± 0.01 e | 0.24 ± 0.01 f |
| Proteins | 7.36 ± 0.06 a | 6.65 ± 0.08 d | 6.27 ± 0.01 e | 5.83 ± 0.02 f | 7.22 ± 0.09 b | 6.86 ± 0.07 c | 5.80 ± 0.01 f | 4.80 ± 0.08 g |
| Ash | 3.42 ± 0.02 a | 2.93 ± 0.08 b | 2.58 ± 0.09 d | 2.38 ± 0.07 ef | 3.02 ± 0.07 b | 2.77 ± 0.06 c | 2.51 ± 0.08 de | 2.25 ± 0.09 f |
| Carbohydrates | 88.75 ± 0.05 f | 90.1 ± 0.1 d | 90.84 ± 0.04 c | 91.54 ± 0.06 b | 89.3 ± 0.1 e | 90.03 ± 0.01 d | 91.41 ± 0.06 b | 92.7 ± 0.1 a |
| Energy (Kcal/100 g dw) | 388.6 ± 0.1 e | 390.1 ± 0.3 d | 391.2 ± 0.3 bc | 391.8 ± 0.2 ab | 390.0 ± 0.2 d | 390.6 ± 0.1 cd | 391.4 ± 0.2 b | 392.2 ± 0.3 a |

**Table 6.** *Cont.*

| | W1 | W2 | W3 | W4 | W5 | W6 | W7 | W8 |
|---|---|---|---|---|---|---|---|---|
| Free sugars | | | | (g/100 g dw) | | | | |
| Fructose | 14.99 ± 0.07 c | 14.97 ± 0.08 c | 15.64 ± 0.02 a | 12.46 ± 0.03 f | 12.72 ± 0.07 e | 15.39 ± 0.06 b | 11.77 ± 0.08 g | 13.76 ± 0.05 d |
| Glucose | 8.54 ± 0.06 a | 7.50 ± 0.08 b | 7.39 ± 0.07 b | 5.34 ± 0.08 g | 6.14 ± 0.07 e | 6.61 ± 0.01 d | 5.88 ± 0.05 f | 7.05 ± 0.01 c |
| Sucrose | 15.77 ± 0.03 e | 14.53 ± 0.08 g | 15.50 ± 0.04 f | 21.85 ± 0.08 a | 21.90 ± 0.04 a | 19.86 ± 0.01 c | 21.24 ± 0.01 b | 19.53 ± 0.01 d |
| Trehalose | 0.27 ± 0.01 d | 0.38 ± 0.01 a | 0.39 ± 0.01 a | 0.30 ± 0.01 c | 0.31 ± 0.03 c | 0.31 ± 0.01 c | 0.30 ± 0.01 c | 0.34 ± 0.01 b |
| Total | 39.6 ± 0.2 e | 37.4 ± 0.2 h | 38.92 ± 0.01 g | 40.0 ± 0.2 d | 41.07 ± 0.01 b | 42.2 ± 0.1 a | 39.19 ± 0.03 f | 40.69 ± 0.07 c |
| Organic acids | | | | (g/100 g dw) | | | | |
| Oxalic acid | 0.067 ± 0.012 d | 0.083 ± 0.001 c | 0.059 ± 0.001 e | 0.050 ± 0.001 f | 0.071 ± 0.001 d | 0.079 ± 0.001 c | 0.098 ± 0.001 a | 0.090 ± 0.001 b |
| Malic acid | 1.39 ± 0.03 d | 1.62 ± 0.02 b | 1.55 ± 0.01 c | 1.75 ± 0.04 a | 1.19 ± 0.02 f | 1.25 ± 0.01 e | 1.16 ± 0.01 g | 1.25 ± 0.01 e |
| Ascorbic acid | tr | tr | tr | tr | tr | tr | tr | tr |
| Citric acid | 1.40 ± 0.01 c | 1.36 ± 0.01 d | 1.43 ± 0.01 b | 1.48 ± 0.01 a | 1.35 ± 0.01 d | 1.24 ± 0.01 e | 1.21 ± 0.01 f | 1.35 ± 0.01 d |
| Fumaric acid | tr | tr | tr | tr | tr | tr | tr | tr |
| Total | 2.86 ± 0.03 c | 3.06 ± 0.03 b | 3.04 ± 0.01 b | 3.28 ± 0.03 a | 2.62 ± 0.02 ef | 2.56 ± 0.01 f | 2.47 ± 0.01 g | 2.69 ± 0.02 d |

Mean values of the same column followed by different letters are significantly different according to Tukey's honest significant difference (HSD) test at *p* < 0.05. The description of treatments is presented in detail in Table 2. tr: traces.

3.3.2. Free Sugars

The main detected sugars in all the tested samples were sucrose and fructose (the values ranged from 14.53 to 21.90 g/100 g dw and from 11.77 to 15.64 g dw, for fructose and sucrose, respectively); followed by glucose (5.34 to 8.54 g/100 g dw) and trehalose (0.27 to 0.39 g/100 g dw) (Table 6). The same main sugars were also detected by Muhammad Jawad et al. [73], who studied free sugar composition during fruit development, while Umer et al. [74] identified specific genes that regulate free sugar biosynthesis during fruit development and maturation. Moreover, Liu et al. [75], who studied several watermelon genotypes, suggested sucrose as the main sugar followed by fructose for seven out of twelve genotypes, whereas for the rest of the genotypes, either fructose was the prevalent sugar (three genotypes in total) or fructose and sucrose were detected in similar amounts (two genotypes in total). However, apart from the genotypic effect, the relative content of the main free sugars in watermelon fruit is associated with maturation stage, since sucrose is transported to fruit via phloem and there it is enzymatically converted to glucose and fructose [74]. On the other hand, a varied effect of the tested formulations was observed. In particular, the highest sucrose content was recorded for the W4 and W5 treatments (21.85 and 21.90 g/100 g dw, respectively), while fructose was the highest for the W4 treatment (15.64 g/100 g dw). Moreover, glucose content was significantly increased in the W1 treatment, while trehalose increased in both W2 and W3 treatments (0.38 and 0.39 g/100 g dw, respectively). Interestingly, the highest total sugar content was detected in the W6 treatment, due to a balanced content of individual free sugars. In contrast to our study, Soteriou et al. [42] did not record a significant effect of protein hydrolysates application on the free sugar content of watermelon fruit, whereas a biostimulant that included polysaccharides, vitamins, amino acids, potassium oxide and polypeptides increased glucose and fructose content in pepper fruit (*Capsicum annuum* L. cv Palermo) [76]. A similar effect was recorded for the application of free amino acids, which increased monosaccharides content in tomato fruit [77], whereas Peris-Felipo et al. [58] suggested that only fructose content was affected by silicon application on strawberry plants. The beneficial effects of biostimulants could be associated with improved biosynthetic processes and the storage of organic carbon surplus in the form of free sugars [78,79]. According to Karagiannis et al. [60], Si application may induce the synthesis of precursor compounds in sugar metabolism, which results in the accumulation of free sugars such as fructose and sorbitol in the flesh and peel of apple fruit. This mechanism is of high importance, especially under stress conditions where sugars and other osmolytes play osmoprotective roles that facilitate crop resilience to stressors [58,80,81]. Moreover, Fernandes et al. [27] reported a significant effect of various biostimulant products that differed in their composition (e.g., yeasts, *Bacillus subtilis* and *Ascophyllum nodosum* extracts; *Glomus* sp.; minerals chelated with glycine), on the sugar composition of tomato fruit.

### 3.3.3. Organic Acids

The composition of organic acids is presented in Table 6, where malic and citric acid were the most abundant compounds, followed by oxalic acid, whereas only traces of ascorbic and fumaric acid were detected. The highest content of individual and total organic acids was observed for the W4 treatment (1.75, 1.48 and 3.28 g/100 g dw for malic, citric and total organic acids, respectively), except for the case of oxalic acid, which was the most abundant in the W7 treatment (0.098 g/100 g dw). It should be noted that these two treatments (e.g., W4 and W7) showed contrasting trends in terms of individual and total organic acid content. The same organic acids were detected in watermelon fruit by Umer et al. [74], who proposed the involvement of specific genes that regulate organic acid biosynthesis and are pivotal for the sensorial quality of fruit. The application of biostimulants may have a varied effect on organic acid accumulation depending on the species and the biostimulant composition [30,82]. Similarly to our study, Hu et al. [83] suggested that Si application increased citric, oxalic and malic acid contents in cucumber fruit, although in our study, oxalic acid content increased only for the W7 treatment, compared to fruit obtained from control plants. This increase of organic acid contents recorded for the W4 treatment in our study could be associated with the low fruit yield for this particular treatment, since according to Weber et al. [82], there is a lower sink–source relationship for plants with lower yields compared to those with better crop performance. Moreover, among the positive effects of Si, an increase in photosynthetic activity, which results in improved biosynthesis of organic compounds such as those in leaves and their translocation to fruit, has been suggested [84].

### 3.3.4. Lipophilic Compounds (Fatty Acids and Tocopherols)

The composition of lipophilic compounds is presented in Table 7. The main fatty acids detected were palmitic acid (C16:0), with contents that ranged between 42.35% and 43.7%, followed by oleic (C18:1n9c) and stearic acid (C18:0), with values that ranged from 14.99% to 15.86% and 13.3% to 14.7%, respectively. Other minor compounds included palmitoleic (C16:1), margaric (C17:0), linoleic (C18:2n6c), arachidic (C20:0), heneicosylic (C21:0), behenic (C22:0), tricosylic (C23:0) and lignoceric acid (C24:0). These results are in agreement with other studies which suggested palmitic acid as the main fatty acid in watermelon fruit, followed by stearic and oleic acids [85,86]. Moreover, the prevailing class of fatty acids was the saturated fatty acids (SFA; 74.37% to 76.2%) due to the high contents of palmitic and stearic acid, while monounsaturated fatty acids were the second most abundant class of fatty acids (17.9% to 18.6%). Regarding the effect of biostimulant formulations, a varied effect was recorded. In particular, palmitic acid content was not significantly different between W4 and W8 treatments, which recorded the highest content, while oleic acid was the highest in the W1 treatment. Finally, oleic acid content did not differ between W1, W2, W4 and W5 treatments. On the other hand, SFA content was the highest for the W7 and W8 treatments, while MUFA and polyunsaturated fatty acids (PUFAs) content did not differ between W3 and W8 treatments and between W1, W3 and W6 treatments, respectively. Our results are in agreement with previous reports on the biostimulant effect on the fatty acid composition of horticultural products, such as tomato [27] or common bean [30]. The positive effects of biostimulant application could be associated with the inhibition of lipid peroxidation through the scavenging of reactive oxygen species, or through the improvement of photosynthetic processes that may result in increased organic carbon pools through the increased content of fatty acids [27,30]. Moreover, considering that the lowest content of PUFA was found for the control treatment, it could be suggested that the tested biostimulant formulations protected fatty acids with more than one double bond from peroxidation; hence the higher relative content detected in fruit from treated plants compared to the control ones [87].

**Table 7.** Chemical composition with regard to lipophilic compounds of the studied watermelon samples (mean ± SD, *n* = 3) in relation to biostimulant formulation.

| | W1 | W2 | W3 | W4 | W5 | W6 | W7 | W8 |
|---|---|---|---|---|---|---|---|---|
| **Fatty Acids** | Relative Percentage (%) | | | | | | | |
| C16:0 | 42.35 ± 0.01 c | 42.7 ± 0.2 bc | 42.4 ± 0.2 c | 43.3 ± 0.3 ab | 43.7 ± 0.1 a | 43.6 ± 0.3 a | 43.7 ± 0.4 a | 43.1 ± 0.8 abc |
| C16:1 | 2.11 ± 0.03 d | 2.38 ± 0.01 c | 2.74 ± 0.04 b | 3.21 ± 0.06 a | 3.1 ± 0.2 a | 3.18 ± 0.01 a | 3.2 ± 0.1 a | 3.2 ± 0.1 a |
| C17:0 | 1.06 ± 0.02 e | 1.18 ± 0.04 cd | 1.12 ± 0.01 de | 1.16 ± 0.06 d | 1.25 ± 0.02 c | 1.19 ± 0.06 cd | 1.55 ± 0.01 b | 1.73 ± 0.07 a |
| C18:0 | 14.3 ± 0.2 ab | 14.3 ± 0.2 ab | 14.2 ± 0.2 b | 14.7 ± 0.2 a | 14.43 ± 0.03 ab | 13.6 ± 0.3 c | 13.3 ± 0.3 c | 13.6 ± 0.2 c |
| C18:1n9c | 15.86 ± 0.08 a | 15.5 ± 0.6 abc | 15.70 ± 0.06 ab | 15.2 ± 0.1 bc | 14.99 ± 0.02 c | 15.2 ± 0.2 bc | 15.2 ± 0.1 bc | 15.4 ± 0.6 abc |
| C18:2n6c | 7.1 ± 0.1 a | 7.34 ± 0.01 a | 7.2 ± 0.1 a | 6.5 ± 0.3 c | 6.6 ± 0.2 bc | 7.05 ± 0.03 ab | 5.6 ± 0.5 d | 5.20 ± 0.04 d |
| C20:0 | 3.68 ± 0.09 cd | 3.86 ± 0.08 ab | 3.79 ± 0.09 bc | 3.51 ± 0.06 e | 3.56 ± 0.04 de | 3.96 ± 0.08 a | 4.00 ± 0.09 a | 3.9 ± 0.1 ab |
| C21:0 | 1.23 ± 0.05 b | 1.18 ± 0.08 bc | 1.47 ± 0.01 a | 1.44 ± 0.01 a | 1.22 ± 0.01 b | 1.03 ± 0.01 e | 1.07 ± 0.02 de | 1.12 ± 0.03 cd |
| C22:0 | 5.38 ± 0.03 b | 5.35 ± 0.03 bc | 5.17 ± 0.03 bc | 5.08 ± 0.01 c | 5.09 ± 0.02 bc | 5.35 ± 0.05 bc | 6.38 ± 0.04 a | 6.4 ± 0.4 a |
| C23:0 | 2.61 ± 0.06 a | 1.72 ± 0.07 c | 1.64 ± 0.05 dde | 1.5 ± 0.2 f | 1.52 ± 0.07 ef | 1.63 ± 0.01 def | 1.13 ± 0.01 g | 1.99 ± 0.07 b |
| C24:0 | 4.25 ± 0.01 cd | 4.47 ± 0.08 bcd | 4.53 ± 0.03 ab | 4.5 ± 0.3 bc | 4.46 ± 0.07 bcd | 4.23 ± 0.02 d | 4.8 ± 0.1 a | 4.3 ± 0.2 bcd |
| SFA | 74.88 ± 0.06 bcd | 74.8 ± 0.5 bcd | 74.37 ± 0.01 d | 75.2 ± 0.4 bc | 75.3 ± 0.4 b | 74.6 ± 0.2 cd | 76.0 ± 0.2 a | 76.2 ± 0.5 a |
| MUFA | 17.98 ± 0.05 b | 17.9 ± 0.6 b | 18.4 ± 0.1 ab | 18.36 ± 0.08 ab | 18.1 ± 0.1 ab | 18.4 ± 0.2 ab | 18.4 ± 0.3 ab | 18.6 ± 0.4 a |
| PUFA | 7.1 ± 0.1 a | 7.34 ± 0.01 a | 7.2 ± 0.1 a | 6.5 ± 0.3 c | 6.6 ± 0.2 bc | 7.05 ± 0.03 ab | 5.6 ± 0.5 d | 5.20 ± 0.04 d |
| **Tocopherols** | (µg/100 g dw) | | | | | | | |
| α-Tocopherol | 120.8 ± 0.6 d | 128.8 ± 0.6 b | 125.2 ± 0.6 c | 136.2 ± 0.3 a | 105.2 ± 0.6 f | 101.0 ± 0.3 g | 83.4 ± 0.3 h | 114.2 ± 0.8 e |

Palmitic acid (C16:0); palmitoleic acid (C16:1); margaric acid (C17:0); stearic acid (C18:0); oleic acid (C18:1n9); linoleic acid (C18:2n6c); arachidic acid (C20:0); heneicosylic acid (C21:0); behenic acid (C22:0); tricosylic acid (C23:0); lignoceric acid (C24:0); SFA: saturated fatty acids; MUFA: monounsaturated fatty acids; PUFA: polyunsaturated fatty acids. Mean values of the same column followed by different letters are significantly different according to Tukey's honest significant difference (HSD) test at *p* < 0.05. The description of treatments is presented in detail in Table 2.

In terms of tocopherols, only α-tocopherol was detected, with the highest amounts being recorded for the W4 treatment, whereas the lowest content was observed for the W7 treatment (Table 7; 136.2 and 83.4 µg/100 g dw, respectively). According to the literature, α-tocopherol is the main vitamin E isoform detected in watermelon fruit [88,89], while small amounts of γ-tocopherol have been also reported [90]. Biostimulant application has been shown to have positive effects on tocopherol composition when plants are grown under stress conditions, since this particular compound is involved in the plant defense system and improves tolerance to stressors [91,92]. However, considering that the plants in our study were grown under non-stressful conditions, the induction of tocopherol biosynthesis needs to be further studied in order to reveal the actual mechanisms involved in this process. A possible explanation could be the beneficial effects of Si on the leaf structure, which becomes more erect and increases its light harvesting efficiency and photosynthetic activity and results in improved biosynthesis of bioactive compounds [93,94].

### 3.3.5. Carotenoid and Chlorophyll Content

The content of pigments (carotenoids and chlorophylls) is presented in Table 8. In terms of carotenoids, β-carotene was detected in amounts that ranged between 0.085 and 0.15 mg/100 g dw, while the highest content was recorded for W2 treatment without being significantly different from the W3 formulation. Similarly, lycopene was also the highest in the W2 treatment (0.14 mg/100 g dw), resulting in the significantly highest total carotenoid content for the same treatment (0.289 mg/100 g dw). The β-carotene and lycopene content detected in our study was below the range reported by Perkins-Veazie [95], although the same authors suggested a significant effect of fruit ripeness and cultivar on the content of carotenoids in watermelon fruit. Moreover, Liu et al. [75] detected not only β-carotene and lycopene but also α- and ζ-carotene and various *cis* isomers of lycopene and β-carotene in lesser amounts in fruit of various watermelon genotypes, while a wide range of content was suggested, indicating a significant genotypic impact on carotenoid composition.

For chlorophyll content, the W1 treatment recorded the highest amounts of individual (e.g., a and b chlorophyll) and total chlorophyll content (0.06, 0.08 and 0.14 mg/100 g dw, respectively), and was significantly different from the rest of the treatments. The high contents of chlorophylls for the W1 treatment are probably associated with the positive effects of Si application on the photosynthetic apparatus of plants, since several studies

have reported the increase of chlorophylls and carotenoids after Si application [41,83,96,97]. However, considering that in most of the studies, Si is applied as a stress alleviation measure that protects chloroplasts from oxidative stress and the fact that all the treatments in our study contained Si, our findings could be correlated with the number of biostimulant applications, suggesting that more than one application might not have a positive effect on plants grown under non-stressful conditions.

**Table 8.** Carotenoid composition (mg/100 g dw) of the studied watermelon samples (mean $\pm$ SD, $n$ = 3) in relation to biostimulant formulation.

| | β-carotene | Lycopene | Total Carotenoids | Chlorophyll a | Chlorophyll b | Total Chlorophylls |
|---|---|---|---|---|---|---|
| W1 | 0.100 ± 0.020 bc | 0.123 ± 0.006 de | 0.225 ± 0.017 c | 0.060 ± 0.020 a | 0.080 ± 0.030 a | 0.14 ± 0.04 a |
| W2 | 0.150 ± 0.010 a | 0.140 ± 0.004 a | 0.289 ± 0.017 a | 0.033 ± 0.009 b | 0.040 ± 0.020 b | 0.07 ± 0.02 b |
| W3 | 0.142 ± 0.008 a | 0.116 ± 0.001 e | 0.258 ± 0.007 b | 0.030 ± 0.020 b | 0.040 ± 0.010 b | 0.07 ± 0.03 b |
| W4 | 0.091 ± 0.009 bc | 0.131 ± 0.004 bc | 0.222 ± 0.005 c | 0.030 ± 0.010 b | 0.040 ± 0.020 b | 0.07 ± 0.02 b |
| W5 | 0.090 ± 0.010 bc | 0.129 ± 0.005 bcd | 0.223 ± 0.007 c | 0.020 ± 0.010 b | 0.040 ± 0.020 b | 0.07 ± 0.01 b |
| W6 | 0.100 ± 0.005 bc | 0.125 ± 0.003 cd | 0.226 ± 0.005 c | 0.018 ± 0.001 b | 0.028 ± 0.001 b | 0.05 ± 0.01 b |
| W7 | 0.106 ± 0.009 b | 0.130 ± 0.002 bcd | 0.236 ± 0.011 c | 0.030 ± 0.010 b | 0.040 ± 0.020 b | 0.07 ± 0.03 b |
| W8 | 0.085 ± 0.009 c | 0.136 ± 0.004 ab | 0.221 ± 0.009 c | 0.020 ± 0.007 b | 0.030 ± 0.010 b | 0.05 ± 0.02 b |

Mean values of the same column followed by different letters are significantly different according to Tukey's honest significant difference (HSD) test at $p < 0.05$. The description of treatments is presented in detail in Table 2.

### 3.4. Bioactive Properties

#### 3.4.1. Antioxidant Activity

The results of the antioxidant activity of the studied extracts are presented in Table 9. The observed activity varied among the treatments, and the lowest $IC_{50}$ values were recorded for the W2, W3 and W8 formulations, without significant differences between them ($IC_{50}$ values of 907, 894 and 902 μg/mL, respectively). However, none of the tested extracts presented higher activity than Trolox, which was used as positive control ($IC_{50}$ value of 21.8 μg/mL), and the control treatment (W8) was among the treatments that presented the highest antioxidant activity and the lowest $IC_{50}$ value overall. To the best of our knowledge, there are no other reports that have determined antioxidant activity in watermelon fruit with an OxHLIA assay to use as a benchmark. In general, Si application is associated with the induction of protective mechanisms of plants against oxidative damage [98,99]; however, considering that all the studied treatments included Si in their composition, this argument does not justify our findings, and other mechanisms have to be postulated focusing on bioactive compounds that may improve the antioxidant capacity of fruit extracts, such as cucurbitacins or citrulline [100,101].

**Table 9.** Antioxidant, cytotoxicity and anti-inflammatory activities of the hydroethanolic extracts of the studied watermelon samples (mean $\pm$ SD, $n = 3$) in relation to biostimulant formulation.

| | W1 | W2 | W3 | W4 | W5 | W6 | W7 | W8 | Positive Controls |
|---|---|---|---|---|---|---|---|---|---|
| Antioxidant activity (IC$_{50}$; µg/mL) | | | | | | | | | Trolox |
| OxHLIA ($\Delta t$ = 60 min) | 1729 $\pm$ 168 d | 907 $\pm$ 65 e | 894 $\pm$ 85 e | 4326 $\pm$ 163 b | 3845 $\pm$ 153 bc | 3738 $\pm$ 102 c | 6435 $\pm$ 393 a | 902 $\pm$ 34 e | 21.8 $\pm$ 0.2 |
| Cytotoxicity to tumor cell lines (GI$_{50}$; µg/mL) | | | | | | | | | Ellipticine |
| AGS | >400 | >400 | >400 | >400 | >400 | >400 | >400 | >400 | 0.90 $\pm$ 0.07 |
| CaCo | >400 | >400 | >400 | >400 | >400 | >400 | >400 | >400 | 0.83 $\pm$ 0.05 |
| MCF-7 | >400 | >400 | >400 | >400 | >400 | >400 | >400 | >400 | 1.02 $\pm$ 0.01 |
| NCI-H460 | 14 $\pm$ 1 f | 112 $\pm$ 9 d | 55 $\pm$ 5 e | 178 $\pm$ 6 bc | 233 $\pm$ 7 a | 166 $\pm$ 7 c | 190 $\pm$ 12 b | 162 $\pm$ 13 c | 1.01 $\pm$ 0.01 |
| Cytotoxicity to non-tumor cell lines (GI$_{50}$; µg/mL) | | | | | | | | | Ellipticine |
| Vero | >400 | >400 | >400 | >400 | >400 | >400 | >400 | >400 | 0.64 $\pm$ 0.05 |
| Anti-inflammatory activity (EC$_{50}$; µg/mL) | | | | | | | | | Dexamethasone |
| RAW 264.7 | 27 $\pm$ 2 c | 31 $\pm$ 2 bc | 30 $\pm$ 2 bc | 32 $\pm$ 3 bc | 35 $\pm$ 3 b | 32 $\pm$ 2 bc | 42 $\pm$ 2 a | 45 $\pm$ 4 a | 16 $\pm$ 1 |

Mean values of the same column followed by different letters are significantly different according to Tukey's honest significant difference (HSD) test at $p < 0.05$. The description of treatments is presented in detail in Table 2.

### 3.4.2. Cytotoxicity

The determination of cytotoxic effects of the tested extracts shows no activity against the tested cell lines (e.g., $GI_{50} > 400$ µg/mL), except for the case of non-small lung cancer cells (NCI-H460 cell line), where all the extracts showed a varied activity (Table 9). In particular, the W1 treatment recorded the lowest $GI_{50}$ values (14 µg/mL), and hence the highest cytotoxicity; however, this activity was lower than ellipticine, which was used as the positive control (1.01 µg/mL). Moreover, none of the extracts showed toxicity against the studied non-tumor cell line (Vero). According to Lu et al. [102], the different parts of watermelon fruit (e.g., flesh, rind and skin) showed a varied efficacy against human intestinal epithelial Caco-2 cells, and they attributed these effects to citrulline content. Moreover, El Gizawy et al. [103] reported varied cytotoxic effects of rind extracts of watermelon fruit against human cell lines including A549, Caco-2, H1299, HCT116, Hep2, HepG2, and MCF-7. In the same study, the authors highlighted the efficacy of aqueous extracts of rinds against HCT116 and Hep2 through suppression of cell proliferation, inhibition of cell migration and cell apoptosis [103]. On the other hand, Cruz et al. [101] did not report any anti-tumor effect for watermelon pulp juice against murine melanoma (B16F10) cells; however, these authors suggested that the administration of these extracts may reduce the toxicity of cisplatin for the treatment of melanoma. Apart from fruit rinds or peels, watermelon seeds are the most studied part in terms of cytotoxic effects, and several studies have reported significant effects against various cell lines due to the presence of various bioactive compounds [104–106].

### 3.4.3. Anti-Inflammatory Activity

All the extracts exhibited significant anti-inflammatory activity, comparable to dexamethasone, which was the positive control of this assay. In particular, most of the tested extracts showed low $EC_{50}$ values, which ranged between 27 and 45 µg/mL, while treatments W1–W4 and W6 recorded the lowest activity, without significant differences between them. According to Itoh et al. [107], two phenolic glycosides (e.g., citrulluside H and citrulluside T) obtained from young watermelon fruit were effective against skin inflammation induced by *Cutibacterium acnes*. In another study, Rafi et al. [108] reported that lycopene may inhibit inflammation in RAW 264.7 cell lines and suggested that fruits that are rich in lycopene could be useful as anti-inflammatory agents. Moreover, protein hydrolysates for watermelon seeds protected RAW 264.7 cells from oxidative damage [109]; while Fetni et al. [110] and Darwish et al. [111] reported similar effects for fruit extracts of *Citrullus colocynthis*. Considering the limited reports regarding the anti-inflammatory activity of watermelon fruit, our results could provide useful information about the effect of simple agronomic techniques, such as biostimulant application, on the bioactive properties of fruit extracts.

### 3.4.4. Antimicrobial Properties
Antibacterial Activity

The antibacterial activity of the hydromethanolic extracts is presented in Table 10. Our results suggested a variable efficacy of the tested samples against the studied bacteria, while in most cases the recorded minimal inhibition concentration (MIC) and minimal bactericidal concentration (MBC) values were higher (less effective) than the positive controls (e.g., E211 and/or E224). The highest efficacy was recorded against *Salmonella typhimurium,* where extracts from W1, W4 and W5 treatments had lower MIC values than both positive controls (0.50 µg/mL compared to 1.00 µg/mL) and similar MBC values to E224 (0.50 µg/mL). Moreover, extracts from W4 and the control treatment (W8) had similar MIC and MBC values to E224 against *Escherichia coli* (0.50 µg/mL), while the extracts from W1 treatment recorded similar MIC values to E224 against *Staphylococcus aureus* (1.00 µg/mL). Most of the existing studies in the literature refer to the antibacterial activities of watermelon seeds and rind, whereas to the best of our knowledge, there are scarce literature reports regarding the antibacterial effects of watermelon flesh. This is probably because usually

pulp extracts show low antimicrobial effects, as indicated by Neglo et al. [112], who suggested that peel extracts had the highest efficacy, followed by seeds, rind and pulp. In another study, alginate biofilms from watermelon seeds modified with melanin silver and zinc oxide nanoparticles were effective against *Escherichia coli* and *Staphylococcus aureus* without showing any cytotoxic effects [113], while El Sayed et al. [114] attributed the antibacterial properties to the presence of various compounds, such as alkaloids, flavonoids, saponins and fatty acids. Moreover, peel extracts can be also effective against various bacterial strains, as was suggested for titanium oxide dots using watermelon peel extracts, which showed promising results against *Bacillus subtilis* and *E. coli* [115]. Both rind and seed extracts of watermelon showed efficacy against various strains of *Staphylococcus* bacteria, *Klebsiella pneumonia*, *E. coli* and *Pseudomonas aeruginosa*, while the seed extracts resulted in higher inhibition than rind extracts [116]. Similarly, ethyl acetate extracts from seeds inhibited the growth of several bacteria, including *S. aureus*, *E. coli*, *Bacillus subtilis*, *P. aeruginosa* and *Candida albicans* [117], while protein fractions from watermelon seeds were effective against *Acinetobacter baumannii* [118]. On the other hand, peel extracts showed antimicrobial potential against *E. coli*, *Staphylococcus epidermidis*, *Trichophyton mentagrophytes* and *Brevibacterium linens* [119–121]. Apart from rinds and seeds, silver nanoparticles from watermelon leaf extracts were also efficient against various bacterial strains such as *S. aureus*, *B. subtilis*, *E. coli* and *P. aeruginosa*, indicating the distribution of valuable bioactive compounds in various parts of the plant [122].

**Table 10.** Antibacterial activity (minimal inhibition concentration (MIC) and minimal bactericidal concentration (MBC) (mg/mL) of the hydroethanolic extracts of the studied watermelon samples in relation to biostimulant formulation.

| | | *S. aureus* (ATCC 11632) | *B. cereus* (Clinical Isolate) | *L. monocytogenes* (NCTC 7973) | *S. typhimurium* (ATCC 13311) | *E. coli* (ATCC 25922) | *E. cloacae* (ATCC 35030) |
|---|---|---|---|---|---|---|---|
| W1 | MIC | 1.00 | 1.00 | 1.00 | 0.50 | 1.00 | 1.00 |
| | MBC | 2.00 | 2.00 | 2.00 | 1.00 | 2.00 | 2.00 |
| W2 | MIC | 4.00 | 1.00 | 2.00 | 1.00 | 1.00 | 1.00 |
| | MBC | 8.00 | 2.00 | 4.00 | 2.00 | 2.00 | 2.00 |
| W3 | MIC | 2.00 | 1.00 | 1.00 | 1.00 | 1.00 | 1.00 |
| | MBC | 4.00 | 2.00 | 2.00 | 2.00 | 2.00 | 2.00 |
| W4 | MIC | 2.00 | 1.00 | 2.00 | 0.50 | 0.50 | 2.00 |
| | MBC | 4.00 | 2.00 | 4.00 | 1.00 | 1.00 | 4.00 |
| W5 | MIC | 2.00 | 1.00 | 2.00 | 0.50 | 1.00 | 1.00 |
| | MBC | 4.00 | 2.00 | 4.00 | 1.00 | 2.00 | 2.00 |
| W6 | MIC | 2.00 | 1.00 | 2.00 | 1.00 | 1.00 | 1.00 |
| | MBC | 4.00 | 2.00 | 4.00 | 2.00 | 2.00 | 2.00 |
| W7 | MIC | 2.00 | 1.00 | 2.00 | 1.00 | 1.00 | 2.00 |
| | MBC | 4.00 | 2.00 | 4.00 | 2.00 | 2.00 | 4.00 |
| W8 | MIC | 4.00 | 1.00 | 2.00 | 1.00 | 0.50 | 2.00 |
| | MBC | 8.00 | 2.00 | 4.00 | 2.00 | 1.00 | 4.00 |
| E211 | MIC | 4.00 | 0.50 | 1.00 | 1.00 | 1.00 | 2.00 |
| | MBC | 4.00 | 0.50 | 2.00 | 2.00 | 2.00 | 4.00 |
| E224 | MIC | 1.00 | 2.00 | 0.50 | 1.00 | 0.50 | 0.50 |
| | MBC | 1.00 | 4.00 | 1.00 | 1.00 | 1.00 | 0.50 |

Antifungal Activity

The studied extracts showed significant antifungal activity, which in several cases was higher than the positive controls used (e.g., E211 and E224) (Table 11). In particular, extracts from W1, W2 and W4 treatments showed higher MIC values than both positive controls and similar MBC values to E224 against *Aspergillus fumigatus* (0.50 and 1.00 mg/mL, respectively), while a similar trend was recorded for W1, W2, W3, W4 and W8 treatments against *A. ochraceus* (MIC and MFC values of 0.50 and 1.00 mg/mL, respectively), for W4 and W5 treatments against *A. niger* (MIC and MFC values of 0.50 and 1.00 mg/mL, respectively),

for W1, W6, W7 and W8 treatments against *Penicillium verrucosum* var. *cyclopium* (MIC and MFC values of 0.50 and 1.00 mg/mL, respectively), and for W1, W3, W4, W7 and W8 treatments against *Trichoderma viride* (MIC and MFC values of 0.25 and 0.50 mg/mL, respectively). Finally, all the extracts had similar MIC values to E224 against *P. funiculosum* (0.50 and 1.00 mg/mL).

**Table 11.** Antifungal activity (MIC and minimal fungicidal concentration (MFC), mg/mL) of the hydroethanolic extracts of the studied watermelon samples in relation to biostimulant formulation.

|  |  | *A. fumigatus* (ATCC 9197) | *A. ochraceus* (ATCC 12066) | *A. niger* (ATCC 6275) | *P. funiculosum* (ATCC 36839) | *P. v. var. cyclopium* (Food Isolate) | *T. viride* (IAM 5061) |
|---|---|---|---|---|---|---|---|
| W1 | MIC | 0.50 | 0.50 | 1.00 | 0.50 | 0.50 | 0.25 |
|  | MFC | 1.00 | 1.00 | 2.00 | 1.00 | 1.00 | 0.50 |
| W2 | MIC | 0.50 | 0.50 | 1.00 | 0.50 | 1.00 | 0.50 |
|  | MFC | 1.00 | 1.00 | 2.00 | 1.00 | 2.00 | 1.00 |
| W3 | MIC | 1.00 | 0.50 | 1.00 | 0.50 | 1.00 | 0.25 |
|  | MFC | 2.00 | 1.00 | 2.00 | 1.00 | 2.00 | 0.50 |
| W4 | MIC | 0.50 | 0.50 | 0.50 | 0.50 | 1.00 | 0.25 |
|  | MFC | 1.00 | 1.00 | 1.00 | 1.00 | 2.00 | 0.50 |
| W5 | MIC | 1.00 | 1.00 | 0.50 | 0.50 | 1.00 | 1.00 |
|  | MFC | 2.00 | 2.00 | 1.00 | 1.00 | 2.00 | 2.00 |
| W6 | MIC | 1.00 | 1.00 | 1.00 | 0.50 | 0.50 | 0.50 |
|  | MFC | 2.00 | 2.00 | 2.00 | 1.00 | 1.00 | 1.00 |
| W7 | MIC | 1.00 | 1.00 | 1.00 | 0.50 | 0.50 | 0.25 |
|  | MFC | 2.00 | 2.00 | 2.00 | 1.00 | 1.00 | 0.50 |
| W8 | MIC | 1.00 | 0.50 | 1.00 | 0.50 | 0.50 | 0.25 |
|  | MFC | 2.00 | 1.00 | 2.00 | 1.00 | 1.00 | 0.50 |
| E211 | MIC | 1.00 | 1.00 | 1.00 | 1.00 | 2.00 | 1.00 |
|  | MFC | 2.00 | 2.00 | 2.00 | 2.00 | 4.00 | 2.00 |
| E224 | MIC | 1.00 | 1.00 | 1.00 | 0.50 | 1.00 | 0.50 |
|  | MFC | 1.00 | 1.00 | 1.00 | 0.50 | 1.00 | 0.50 |

Similarly to our study, Sathya and Shoba [18] suggested that seed extracts were more efficient against fungi than bacteria, as observed in our study for pulp extracts. Moreover, according to Ali et al. [115], titanium oxide dots using watermelon peel extracts were effective against *Cryptococcus neoformans*, *Candida albicans*, *Aspergillus niger* and *A. fumigatus*, while Aktepe and Baran [122], El Gizawy et al. [103] and Zahng et al. [123] reported that silver nanoparticles of watermelon leaf extracts and rind extracts were effective against *Candida albicans*. Considering the lack of studies on the antifungal effects of extracts obtained from watermelon fruit and the significant activities presented in our results for fruit flesh, it should be highlighted that further research is needed, with different extraction protocols and biostimulants with varied compositions to reveal any beneficial effects on bioactivities that could be valorized in the food industry through the substitution of synthetic antimicrobial agents with naturally-derived ones.

*3.5. Principal Component Analysis*

The principal component analysis (PCA) was performed of complex multivariate data aiming to identify groups and highlight similarities and differences between the tested treatments. Our analysis showed that the first six principal components (PCs) were associated with eigenvalues higher than 1 and explained 94.74% of the cumulative variance, with PC1 accounting for 43.4%, PC2 for 18.8%, PC3 for 15.1%, PC4 for 7.9%, PC5 for 6.0 and PC6 for 3.5%. In particular, PC1 was positively correlated with α-tocopherol, stearic acid, heneicosylic acid, citric acid, malic acid and total organic acids, whereas it was negatively correlated with palmitoleic acid, margaric acid, arachidic acid, behenic acid (C22:0), MUFA, oxalic acid and anti-inflammatory activity. On the other hand, PC2 was positively correlated

with ash, proteins and fat content, palmitic acid and oxalic acid, whereas it was negatively correlated with α-tocopherol, oleic acid, heneicosylic acid, carbohydrate content, citric acid, malic acid, total organic acid and energy content. Finally, PC3 was positively correlated with margaric acid, behenic acid, tricosylic acid, SFA, carbohydrate content, glucose and anti-inflammatory activity, whereas a negative correlation was observed for palmitic acid, linoleic acid, fat and protein content, sucrose, total sugars, NCI-H460, OxHLIA and PUFA. Therefore, PCA facilitates the discrimination of the tested biostimulant formulations, as presented in the respective scatterplots and loading plots. The scatterplot in Figure 2 shows five distinct groups of the tested formulations based on total fruit yield and proximal and chemical composition, and bioactive properties fruit extracts. PC1 discriminated the W1 formulation due to a high content of stearic and oleic acids, as well due to low MUFA content; the W4 formulation due to a high content of malic acid, citric acid, total organic acids, stearic acid and α-tocopherol and the high anti-inflammatory activity, as well as due to a low content of oxalic acid (Figures 3 and 4). PC2 discriminated W1 due to a high content of fat, protein and ash and low carbohydrate, palmitic acid and energy content; W5 due to a high content of palmitic acid (Figures 3 and 4). Finally, PC3 discriminated W1 due to a high content of glucose and the high toxicity against NCI-H460; W5 due to low toxicity against NCI-H460 (Figures 3 and 4).

Moreover, the loading plot of the first two components also revealed groups of positively correlated variables (Figure 3). In the upper left quadrant were included oxalic acid, total sugars, OxHLIA, SFA, RAW 264.7, C16:0, C17:0, C20:0, C22:0; in the lower left quadrant were included NCI-H460, sucrose, C16:1, C24:0 MUFA, yield, carbohydrates and energy; in the upper right quadrant were included fat, ash, glucose, fructose, PUFA, C18:1n9c, C18:2n6c; in the lower right quadrant were included trehalose, α-tocopherol, malic acid, citric acid, total organic acids, C18:0 and C21:0.

The loading plot of PC1 and PC3 correlated variables as follows: in the upper left quadrant were included oxalic acid, carbohydrates, energy, RAW 264.7, C17:0, C20:0, C22:0, MUFA and SFA; in the lower left quadrant were included yield, total sugars, sucrose, NCI-H460, OxHLIA, C16:0, C16:1 and C24:0; in the upper right quadrant were included trehalose, glucose, fructose, α-tocopherol, malic acid, total organic acids, C18:1n9c and C23:0; in the lower right quadrant were included citric acid, fat, ash, proteins, C21:0, C18:2n6c and PUFA (Figure 4).

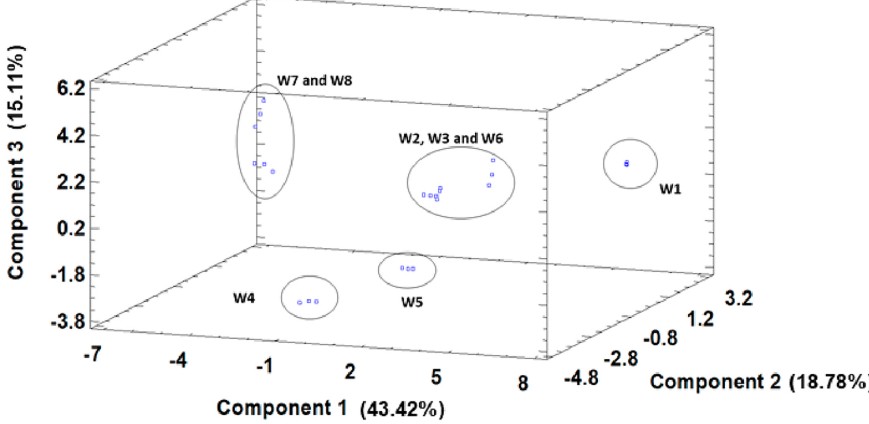

**Figure 2.** Three-dimensional scatterplot of principal components 1, 2, and 3 for the tested biostimulant formulations.

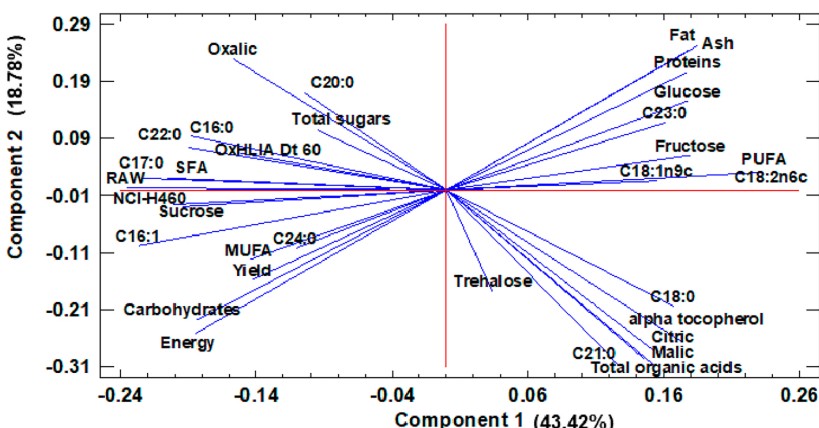

**Figure 3.** The loading plot of principal components 1 and 2 for the tested biostimulant formulations.

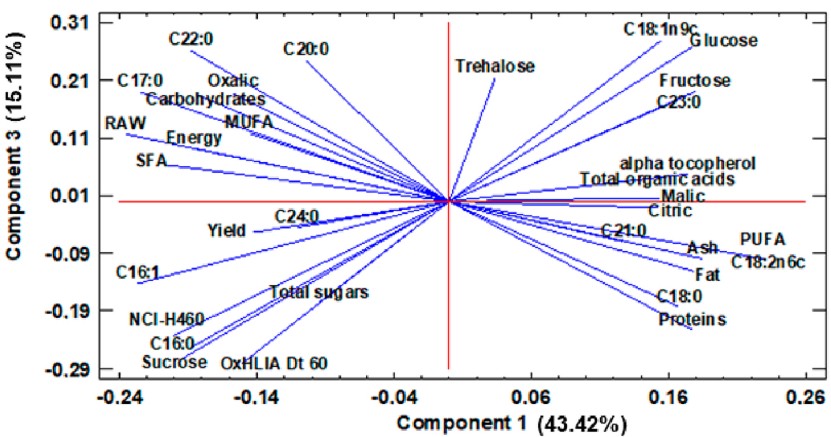

**Figure 4.** The loading plot of principal components 1 and 3 for the tested biostimulant formulations.

## 4. Conclusions

The results of our study highlight the significant effect of biostimulant application on crop performance, proximal composition and the bioactive properties of watermelon fruit. However, a varied effect was recorded depending on the tested formulation, which indicates that although Si application may have a beneficial effect, the number of applications is also important to increase the crop yield and quality of watermelon. Moreover, a variable efficacy of the tested extracts against the studied bacteria and fungi was recorded, while antifungal activity was higher than the extracts compared to the positive controls. Therefore, it could be concluded that simple agronomic practices, such as the application of biostimulants that contain different amounts of Si may improve crop performance and improve the proximal composition and the overall quality of watermelon fruit within the context of sustainable crop production. However, further studies are needed to reveal the possible mechanism of the beneficial effects of the tested formulation and associate them with the number of applications and the application regime.

**Author Contributions:** Conceptualization, S.A.P.; methodology, Â.F., N.P., F.M., C.P. and J.P.; software, Â.F., N.P., F.M., C.P. and J.P.; formal analysis, Â.F., N.P., F.M., C.P. and J.P.; investigation, Â.F., N.P., F.M., C.P., J.P. and M.S.; resources, M.S. and S.A.P.; data curation, Â.F., N.P., J.P., M.S. and S.A.P.; writing—original draft preparation, Â.F., N.P. and S.A.P.; writing—review and editing, M.S. and S.A.P.; visualization, S.A.P.; supervision, M.S. and S.A.P.; project administration, M.S. and S.A.P.;



funding acquisition, M.S. and S.A.P. All authors have read and agreed to the published version of the manuscript.

**Funding:** This research was funded by Agrology S.A., grant number 6089. The authors are grateful to the Foundation for Science and Technology (FCT, Portugal) for financial support through national funds FCT/MCTES to the CIMO (UIDB/00690/2020 and UIDP/00690/2020) and SusTEC (LA/P/0007/2020). In addition, national funding was received by F.C.T., P.I., through the institutional scientific employment program contract for Â.F. This work is funded by the European Regional Development Fund (ERDF) through the Regional Operational Program North 2020, within the scope of Project GreenHealth (Norte-01-0145-FEDER-000042).

**Data Availability Statement:** Data available upon request.

**Acknowledgments:** The authors are grateful to Spyridon Souipas and Sofia Simopoulou for technical assistance throughout the experiment. The authors are also grateful to Agrology S.A. (Greece) for the provision of biostimulant formulations.

**Conflicts of Interest:** The authors declare no conflict of interest.

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
