# Peer review of "Combined Effect of Biostimulants and Mineral Fertilizers on Crop Performance and Fruit Quality of Watermelon Plants"

_horticulturae, doi:10.3390/horticulturae9070838_

Round 1
Reviewer 1 Report
The manuscript presents scientific merit and interest to the reader. My suggestions are in the attached manuscript. Thanks in advance for the opportunity to review it.

The manuscript presents scientific merit and interest to the reader. My suggestions are in the attached manuscript. Thanks in advance for the opportunity to review it.
Reviewer 2 Report
The authors have prepared a manuscript concerning “The Combined Effect of Biostimulants and Mineral Fertilizers on Crop Performance and Fruit Quality of Watermelon Plants Grown in Greece”. I think some improvements are needed. Overall, the manuscript will meet the publishing standard of the journal after revisions.
Abstract: The description of the abstract part of the article is some longer, and it is recommended to reduce it to about 200 words.
Introduction: It is suggested that the first two paragraphs be merged together and that the description of the role of watermelon in the first paragraph be streamlined.
Table: The number of decimal places for the data in the table should be consistent and retained to two places.
Table 4: It is recommended that the specific meaning of flesh color parameters such as “L*, a*” be indicated at the bottom of the table.
Table 6: It is suggested that the specific meaning of "tr" be indicated at the bottom of the table.
Figure 1-3: It is recommended that the contribution rate (%) of the corresponding principal components be labeled in the figure, such as “Component 1 (43.4%)”.
Conclusions: The conclusion section should be concise and clear, and it is recommended that it be condensed to about 150 words.

Reviewer 3 Report
1. Manuscript title: Is this necessary to highlight "grown in Greece"? "The Combined Effect" changes to "Combined effects".
2. Abstract: The suggestion is to concentrate on it, and only describe the most important findings.
3. L39: It's so strange because an abstract itself is a conclusion.
4. L144-146: How many samples were conducted for evaluating soil compositions? It should be reliable by taking enough replicates.
5. Table 1: The compositions of "mixture of natural metabolic catalysts", "amino acids", and "trace elements" should be more defined.
6. Data: The number of decimal places should be unified for all tables, in the current version, zero, two, or three decimal places can be found in the same table.
7. The authors should provide iconic photos of plants to present the overall performance at least for the best treatments to compare with the control plants.
8. It's difficult to understand the logic of the experimental design in Table 2.
9. In my opinion, this study is very difficult to be repeated due to many undefined factors.
Round 2
Reviewer 3 Report
I don't have further questions.
Author Response
We are thankful to the reviewer for accepting our responses to the comments of the previous round.